# Are non-fungible payments attractive when they reduce risk exposure? Evidence from Colombia

Alexander Cano[1], Darwin Cortés[2], César Mantilla[2¤]*, Laura Prada[2], Medardo Restrepo[3]

**1** Justus-Liebig-Universität Giessen, Giessen, Germany, **2** Economics Department, Universidad del Rosario, Bogotá, Colombia, **3** Universidad del Quindío, Armenia, Colombia

¤ Current address: Loyola Behavioral Lab, Universidad Loyola Andalucía, Seville, Spain
* ca.mantilla@uloyola.es

**Data Availability Statement:** The data and code to replicate all the tables and figures can be found at the Open Science Framework platform: https://osf.io/rp4xa/.

## Abstract

We conducted a lab-in-the-field experiment in which 214 Colombian rural workers must choose between cash or voucher payment for completing a real effort task. Although the voucher may be perceived as non-fungible, it halves the probability of suffering a negative shock that will reduce the participant's payment by two-thirds. Participants made four decisions in which we vary the voucher values such that this payment method offers, in expectation, between 88% to 123% of the cash payment (fixed across decisions). We find that uptake rates go from 32% to 56%, from the least to the most generous voucher. These rates are consistently larger compared to a reference sample of undergrad students from the same region ($p$–values from the $\chi^2$ tests for all four decisions fall below 0.035). Our between-subjects variations reveal that presenting the vouchers in descending order yields a higher uptake than the ascending order ($p < 0.001$ for the corresponding coefficient in a tobit and ordered logit regressions including municipality characteristics, an effect driven by the two decisions with the lowest voucher values, with $p$–values of 0.008 and 0.072 in the $\chi^2$ tests, respectively). We interpret this result as an endowment effect of the voucher's risk reduction.

## Introduction

According to the fungibility assumption, units of money are substitutable, and consequently, the consumption decisions should be based on total wealth or income [1]. However, violations of fungibility are well-studied and can be explained by at least three augmented decision-making models incorporating psychological biases: category budgeting (or mental accounting), salience, and loss-aversion [2, 3]. In the context of developing countries, such violations are reported for politicians' public spending decisions [4], cash payouts not affecting supplied effort [5], and intertemporal preferences for earned and unearned (i.e., a windfall of money) income [6].

In rural settings, non-fungibility may interact with the non-separability of farmers' production and consumption decisions [7, 8], with two potential implications: it may halt

**Funding:** DC, CM and MR received funding from MinCiencias and the World Bank through the program "Inclusión productiva y social: programas y políticas para la promoción de una economía formal, código 60185, Alianza EFI". Grant number: FP44842-220-2018. The funders had no role in study design, data collection and analysis, decision to publish, or preparation of the manuscript.

**Competing interests:** The authors have declared that no competing interests exist.

technological adoption and prevent risk-sharing arrangements relying on in-kind transfers. An example of the former would be the undervaluation of agricultural inputs subsidized through voucher transfers or payments. The latter is the purpose of study in this paper, as the following thought experiment suggests.

Imagine that a cooperative, or any alternative community-based institution with aggregation and coordination capabilities, asks members to sell their agricultural output to the cooperative, and part of their payment is given as redeemable vouchers throughout the same cooperative, primarily by purchasing other members' products. If the voucher is fungible and its total value is non-distortionary (i.e., relatively small with respect to their total expenditure), farmers could incorporate the vouchers as part of their food expenditure without altering the total allocation for this item, while simultaneously benefiting from the risk-reduction provided through the cooperative's arrangement that purchases their harvest. This arrangement would work as a risk-pooling mechanism whose cost should be small if fungibility holds. By contrast, if farmers treat vouchers as non-fungible, the payment schemes based on risk-pooling and a partial in-kind payment are unlikely to work as an alternative for strengthening multi-product cooperatives in areas where markets are poorly connected.

We explore these preferences for accepting an in-kind payment in exchange for reducing risk exposure in a lab-in-the-field experiment with Colombian rural workers. Risk reduction may be appealing in rural contexts, where shocks have more persistent effects on health and educational outcomes given the limited means to self-insure [9, 10], and the prevalence of unintended effects after the efforts to increase the supply of health services (e.g., see a review for sub-Saharan Africa in Kolié et al. [11]).

In our setting, participants make four decisions between a cash payment and a voucher immediately redeemable in a nearby grocery store. Choosing the voucher brings a reduction, from 20% to 10%, in the chance of suffering a negative income shock that would deduct two-thirds of the earnings from a real effort task. The cash payment is the same across the four decisions, but we vary the voucher payment. We start with a voucher whose expected value is lower than the cash expected payment. This decision helps understand whether the risk reduction is sufficiently appealing and whether participants are willing to choose this payment method, implicitly revealing that they treat the voucher as fungible. As the voucher's expected value increases in the subsequent decisions, the expected utility model falls short in explaining why participants do not opt for the less risky and better-paid option, revealing that the violations of fungibility make the voucher payment unattractive for some participants.

We conducted this study with 214 rural workers from the coffee growing region of Quindío. The sessions took place in eleven municipalities involved in the study. We typically conducted two sessions per municipality, allowing us to use between-session random variations to explore whether the transaction costs of voucher redemption affect the attractiveness of this payment method. All participants received a show-up fee for participation, but we randomized whether they received it in cash or as a voucher. If participants perceive that the voucher has a positive redemption cost, those receiving the show-up fee in a voucher would be more likely to select this payment for completing the real-effort task because they already need to incur this cost.

We also randomized, at the municipality level, the order in which the four decisions were presented in each session. Recall that, whereas the cash payment was fixed, we employed four different voucher values that changed across decisions. We introduced them in ascending order to shed light on why the expected utility model fell short in explaining behavior, giving room to non-fungibility. However, in half of our sessions, the four decisions were explained to (and collected from) the participants in descending order. This was done to avoid a

confounding problem between the attractiveness of higher voucher values and the increasing attractiveness of moving away from a reference point (i.e., the lowest voucher value).

As a preview of our findings, the voucher uptake went from 32% for the least generous voucher to 56% for the most generous voucher. We did not find differences in the voucher uptake depending on whether the show-up fee was delivered in cash or as a voucher. By contrast, we found weak evidence that presenting the vouchers in descending order (i.e., showing in the first decision the most generous voucher) increased the voucher uptake. We conjecture that, by facing the most attractive voucher first, participants experienced an "endowment effect" [12] with the insurance (i.e., that they place more value on the insurance by the fact of having it), increasing the cash premium required to forgo the lower probability of a loss.

To put the voucher acceptability in perspective, we implemented the same instrument with a sample of 69 university students from the region's main city, Armenia. Non-fungibility should be even starker in this sample since students in this region are rarely in charge of grocery shopping at their homes. Indeed, we found that their voucher uptake went from 16% to 33%, from the least to the most generous voucher. These numbers are considerably below the voucher uptake in the rural sample (e.g., the workers' uptake for the least generous voucher matches the students' uptake for the most generous voucher). Unlike rural workers, students appear to be influenced by the redemption costs: they were more likely to select the voucher if their show-up fee was delivered as a voucher as well.

These results contribute to the literature reporting violations of fungibility in the developing-country context [5, 6], and also to the literature exploring the adoption of risk-pooling mechanisms through the use of lab-in-the-field experiments. The literature devoted to the measurement of risk preferences and its correlation with relevant development outcomes is ample [13–18]. Here, our contribution is measuring the trade-off between a cash payment and accepting an in-kind (potentially non-fungible) payment method that brings a risk-reduction framed as an insurance. Offering this abstract version of an insurance also lets us block a potential problem in which participants associate insurance products with financial institutions. Otherwise, a lower uptake of the risk-reducing choice may be explained by the participant's past experiences or mistrust in these institutions. Our study also sheds light on non-fungibility as one mechanism explaining why cooperatives' efforts to promote risk-pooling agreements may fail when they involve in-kind payments. This list is already ample, and it includes a lack of guarantees for farmers' long-term market participation, the existence of social sanctions, and the absence of enforcement mechanisms [19–21]. The mild attractiveness of non-fungible payments in rural contexts also contributes to the general understanding of the dynamics of rural labor markets, including income diversification in non-farm enterprises (e.g., small informal businesses [22]) and the increase in the household's labor supply to cover unanticipated shocks and anticipated liquidity shortages [23, 24]. This supply response may also include child labor (that the household might see as intergenerational transmission of skills and values, according to qualitative information [25]).

## Experiment

### Experimental design

**Set up of decisions between cash and voucher payment.**   We employed a multiple price list setting that worked as follows. Upon arrival, we informed participants that they would need to perform a real-effort task and that they would get paid in Colombian pesos (COP) for its completion. The average exchange rate between July and August 2021, the months in which fieldwork took place, was 3,877 COP for 1 USD. We also informed them that, at the end of the activity, they could lose a considerable amount of their individual earnings due to a negative

shock. They had to pick one of two choices, combining a payment method and an absolute risk exposure:

- *Cash*: Receive 30 kCOP (7.74 USD) in cash for task completion. Face a **20%** chance of losing 20 kCOP (5.16 USD).

- *Voucher*: Receive $V$ kCOP in a voucher for task completion. Face a **10%** chance of losing 20 kCOP.

We opted for reducing the risk exposure rather than for its entire elimination because the lack of uncertainty may be too appealing for the participants. This alternative design could raise voucher uptake not as a validation of fungibility but instead as a signal of a strong preference for certain outcomes.

Participants took four decisions with different voucher values given by $V \in \{25; 28; 30; 34\}$ [kCOP] (in USD, $V$ goes from 6.45 to 8.77). Our reference point for creating the trade-offs between payment methods was the cash payment of 30 kCOP. This value corresponded to the standard daily wage of rural work (*jornal*) in the time and region of our study. This amount was attractive and compensated for the participants' opportunity cost of time.

To explain better the rationale behind the specific values of $V$, let us introduce the expected values of the cash and voucher payments in Table 1. By design, $E$(Voucher), the expected value of the voucher, is increasing in $V$. By contrast, and to make the decisions more tractable, the cash payment and its expected value, $E$(Cash), are held constant. Our setting is simple compared to the popular risk elicitation method that detects the acceptable binomial probability to switch from a low yield-low risk to a high yield-high risk lottery [26]. Despite the good properties of this method in terms of parameterization of risk preferences, there is evidence from developed and developing countries that it may be confusing among non-student populations [27, 28].

The three rows in the middle of Table 1 provide a direct comparison between the expected payoffs: their difference, their ratio, and their ratio ignoring the potential losses (i.e., the ratio before the chance of losing 20 KCOP).

Let us start with $V = 25$, the lowest voucher value and, therefore, the least attractive with respect to the cash payment. In terms of predictions, it should only be selected by participants that are sufficiently risk averse to value the reduction in risk exposure *and* see the voucher as fungible. Our choice of $V = 25$, instead of any other $V < 28$ that also yielded $E$(Voucher)/$E$(Cash) $< 1$, considered the transaction costs of lowering risk exposure. In fact, the payment ratio 25/30 (or 83% in the table) resembles the contribution rates to social security in the Colombian labor market, which are about 17% of the income. An alternative interpretation is

**Table 1. Comparison, in expected values, between the *Cash* and *Voucher* payment methods.**

| | Voucher value (in kCOP) | | | |
| --- | --- | --- | --- | --- |
| | $V = 25$ | $V = 28$ | $V = 30$ | $V = 34$ |
| $E$(Voucher) | 23 | 26 | 28 | 32 |
| $E$(Cash) | 26 | 26 | 26 | 26 |
| $E$(Voucher) − $E$(Cash) | -3 | 0 | 2 | 6 |
| $E$(Voucher)/$E$(Cash) | 0.88 | 1 | 1.07 | 1.23 |
| Payment Voucher/Cash (pre-shock) | 0.83 | 0.93 | 1 | 1.13 |
| **Predictions under expected utility** | | | | |
| Risk-neutrality and fungibility | Cash | Indifference | Voucher | Voucher |
| Slight risk-aversion and fungibility | Cash | Voucher | Voucher | Voucher |

that, if the risk exposure is directly covered by rural cooperatives, this could be a transaction fee charged by the cooperative for managing this additional risk. More generally, this scenario yields an idea of the acceptance rate of this risk-reduction policy if the users pay the costs associated with the risk transfer as an insurance premium.

When $V = 28$, we have $E$(Voucher)$/E$(Cash) $= 1$. Having the same expected value, and a lower risk exposure with the voucher payment, any degree of risk aversion would be sufficient to prefer the voucher if the payment methods are treated as fungible. Given the prevalence of risk-aversion [29], low to intermediate uptake levels of the voucher payments for $V = 28$ would be more likely to signal the presence of non-fungibility than a disproportionate number of risk-seeking participants.

By setting $V = 30$ in one of our decisions, we can study the choice of payment method while keeping the same initial paid amount (i.e., before the potential loss). This decision is illustrative because it makes the comparison of expected losses very straightforward: both choices pay the same and face a loss of the same magnitude, but the probability of this loss is twice with one payment method with respect to the other. Hence, the trade-off between risk exposure and non-fungibility is much clearer. Choosing the cash over the voucher payment would be, in this case, a violation of fungibility. As an alternative but non-mutually exclusive interpretation, the difference in payment methods would be sufficiently salient to induce a problem of probability neglect [30], where participants are not sensitive to the absolute risk reduction to avoid the non-fungible payment.

Finally, the decision with $V = 34$ yields a larger voucher payment, both *ex ante* (13%) and in expected value (23%), compared to cash. We set up this relatively large $V$ aiming to find a reasonable subsidy amount that would guarantee almost complete acceptability of these non-fungible payments. A voucher acceptance rate far from 100% would suggest that even if the risk-transfer mechanism is externally funded, non-fungibility remains a threat to its popularity.

The bottom of Table 1 summarizes the predictions under the expected utility model for each $V$. Note that, if risk neutrality and fungibility hold, we should observe the choice of cash payments only for $V \leq 28$. If risk neutrality is replaced by mild risk aversion (to break the indifference from a lower risk exposure with the same expected value), we should observe cash payments only for $V = 25$. In the discussion section, we will analyze our findings in the light of alternative behavior models.

**Voucher redemption.** The payment, in vouchers or cash, was delivered to each participant once the session finished. The voucher was a signed paper card indicating the total payment to redeem in a supermarket. The first time we mentioned the vouchers in the protocol, we explained to participants the following redemption procedure. They would be guided by the research assistants to a supermarket located within walking distance from the place where the session took place. Most of (if not all) the participants recognized the name of the supermarket, since the municipalities' capital (or *cabecera*, which the reader may visualize as the main town, though hereafter we will refer to this more aggregated area of the municipality as its capital) tend to be relatively small in extension. They had 90 minutes after the end of the session to redeem the voucher value for products, and they could request the help of the research assistants, who stayed in the supermarkets until the last participant of each session completed her voucher redemption.

By making voucher redemption instantaneous, we focus on instantaneous comparisons between payment methods, blocking inter-temporal mechanisms that have been explored in field and lab-in-the-field experiments [6, 31, 32]. Moreover, once an inter-temporal decision window is open, it becomes harder to disentangle fungibility from liquidity. On the other hand, an attentive reader may wonder whether the *instantaneous* character of voucher redemption is not an additional attribute that could make participants opt for the cash

payment. We argue that this confounding issue was very unlikely because we conducted the sessions during the weekends, the days in which the rural workers visit the municipality's capital to receive their wages and make their purchases. Hence, if fungibility holds, the voucher could perfectly substitute the cash expenditures for food and cleaning products initially planned to be made that same day. Moreover, the wages are delivered weekly, biweekly, or monthly, and the payments in our experiment were equivalent to about one day of salary. Hence, our payments were relatively small and non-distortionary with respect to the income received that day.

**Between-subjects randomizations.** We varied two dimensions of the experiment at the session level: the payment method of the show-up fee and the ordering in which the four decisions were presented to participants.

At the beginning of each session, we informed participants that just for coming and staying until the end of the activity, they would receive 10 kCOP. This show-up fee offers a balance between a fixed payment sufficiently low to preserve the salience of the decision-based payments but sufficiently high to guarantee a minimum compensation of 15 kCOP for participants facing the negative shock. In half of the sessions, this payment was provided as an additional voucher, and in the other half in cash. The logic behind this variation is to detect whether transaction costs from voucher redemption reduce their attractiveness. Imagine that, even if participants planned to make a food expenditure during this visit to the municipality's capital, they often do it elsewhere. Going to a different store may create some transaction costs that, if they are perceived as non-negligible, could make vouchers less attractive. If the transaction costs are non-negligible, we expect participants receiving the show-up fee in vouchers would also be more willing to accept vouchers as a payment method, as they already need to visit the supermarket to redeem their payment.

We also varied the display order of the four decisions, aiming to control for order effects. [2] remarked that three "alternative" models (i.e., capturing a psychological bias) could explain violations of fungibility: category budgeting, salience models, and loss aversion. In the latter, reference-dependence plays an important role that we aimed to capture: since the cash payment was fixed for simplicity, vouchers may become more or less attractive depending on the vouchers previously offered. We asked participants to mark their choice in decision cards as shown in the left panel of Fig 1. In roughly half of our sessions, the decision cards were presented and explained in ascending order (i.e., $V$ foes from 25 to 34). In the remaining sessions,

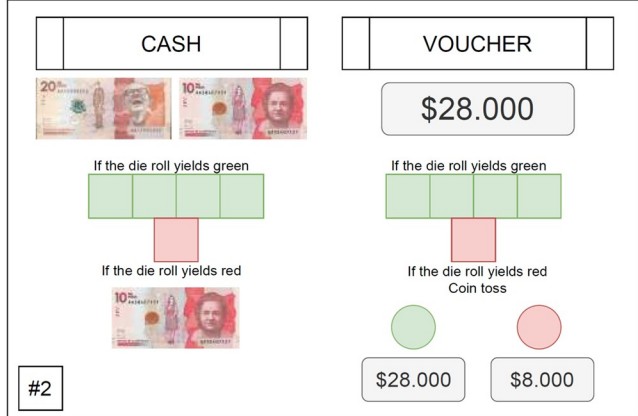
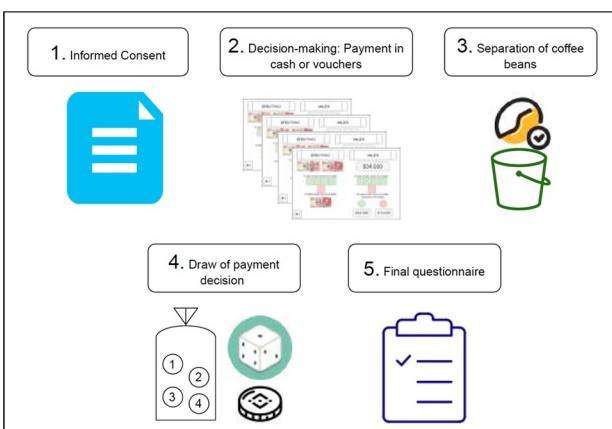

**Fig 1. Experiment materials.** Translated from the original version in Spanish. Left panel: plastic card for a choice between cash and voucher. Right panel: poster with the summary of instructions for participants.

we employ a descending order (i.e., *V* foes from 34 to 25). Moreover, we instructed participants to wait until the experimenter gave the oral instructions for each decision card before making a choice. With this procedure, we aimed to control the pace of the decision-making process and mitigate noisy responses.

We argue that reference-dependent preferences may have two different effects on the sequential choice of voucher or cash payments. If the voucher value creates the reference point, we would expect that the ascending order yields a higher voucher uptake than the descending order. By contrast, if the reference point is created by having the insurance, then loss aversion may trigger an endowment effect that yields a higher voucher uptake for the descending order because it is harder for participants to forgo the lower risk exposure from the vouchers.

## Implementation procedure

The experimental protocol was approved by the Ethics Committee at Universidad del Rosario (approval code 459-CS295). We obtained written consent from all participants after all the instructions were provided. No minors were included in our study. We informed the participants that the study was part of a project about the functioning of labor markets, funded by the Colombian Ministry of Science and implemented by a team of researchers from Universidad del Rosario and Universidad del Quindío. We emphasized our interest in understanding their preferences for cash or voucher payments for completing a task. We also informed participants that the collected information would be only used for academic purposes and that their payment would be made privately, minimizing any potential risk from the study. Our protocol also included a set of rules for conducting in-person sessions during the pandemic. We interacted in ventilated areas, with a maximum of fifteen participants, and mandatory use of hygienic face masks for participants and the research team during the entire activity.

We planned sessions with 10 to 12 participants, although some sessions had a lower attendance. Upon arrival, participants drew a numbered token that defined where they would be seated. They were instructed to keep this token until the end so we could guarantee their anonymity in the payment by calling them by their assigned number.

**Timing of a session.** *(i) Introduction:* We started with the general instructions and explained how the show-up fee would operate, depending on their treatment. We rapidly introduced the real-effort task, so the participants got an idea of why they were getting paid. Then, we explained in detail their payment options for completing the task: either payment in vouchers or cash. We made a detailed description of the cash and voucher options, emphasizing two aspects: the voucher provided partial insurance against the chances of losing 20 kCOP (about two-thirds) from their total earnings, and it was redeemable in a specific supermarket during a limited time, after the activity. We mentioned the name of the supermarket to make sure that all the participants knew the store's location.

*(ii) Introduction to the four decisions:* We explained to participants that they would make four decisions, but only one would be used to compute their payments. To do so, they knew they would have to draw a ball from an urn containing four balls numbered 1 to 4. We made clear that the payment in cash would be the same across decisions but that we were interested in knowing "the payment in vouchers that would make this option attractive." Therefore, the voucher value in each decision would be different.

*(iii) Introduction to the real-effort task:* Participants received a pot containing white and red beans and a smaller empty pot. The task, emulating manual chores involving coffee beans, was to separate the kidney beans by moving the red ones to the smaller pot. We clarified that earnings depend on task completion, regardless of the time taken to separate the beans by color.

We also informed participants that, although it does not affect payments, we will measure the time taken to finish this task.

*(iv) Detailed explanation of decisions:* For each decision, participants received a large plastic card where they were asked to mark with an "X" their preferred choice between the cash and the voucher payment. With this choice method, we aimed to minimize the participants' requirements in terms of writing and reading to avoid excluding less educated participants from our study. The left panel in Fig 1 displays an example of the decision card, which graphically shows the probabilities and consequences of having a negative shock.

We explained to participants that, to compute their final payment, each one of them will need to roll a die having four green sides, one red side, and one blank side. The blank side would indicate a die re-roll. Any of the green sides would guarantee the full payment, and falling into the red side would imply a loss of 20 kCOP, our shock. However, if the participant chooses the voucher, she will also toss a coin with one green and one red side. If the toss outcome were green, the "insurance" from the voucher would make the participant avoid the 20 kCOP loss. Only the red-red combination in the die and the coin would cause a loss to a player getting paid with the voucher, whereas a red outcome in the die was sufficient to cause a loss in case of a cash payment.

*(v) Getting consent:* After receiving these instructions, participants signed the informed consent.

*(vi) Decision-making:* For each decision, we gave each participant the plastic card corresponding to the current decision. We explained in detail the payoffs for the decision and reminded them that they needed to mark with an "X" their preferred choice. We repeated this step, including the payoff explanation, for each decision. Participants were not allowed to mark the decision cards until we provided the specific explanation of each decision's payoffs.

*(vii) Completion of the real-effort task:* Participants took the two pots and separated the beans. We registered how much time it took them, but reminded them that incentives depended on completion and not on time taken.

*(viii) Draw of payment decision:* A field assistant approached each participant and asked her to draw a numbered ball out of four, dictating the task chosen for payment. Then, each participant rolled the die and tossed the coin. We asked all participants to toss the coin, regardless of their payment choices, for two reasons: to keep anonymity in the decisions and to speed up the payoff resolution phase.

*(ix) Questionnaire:* We administered a survey to collect information regarding the participant's demographics, access to credit, and response to health shocks. This survey included a non-incentivized measurement of willingness to take risks by Dohmen et al. [33], who showed that responses are correlated with the choice of more risky incentivized lotteries. We employed this non-incentivized question to reduce the cognitive burden from our instrument and avoid further steps in the payoff computations, which may confuse participants.

The right panel of Fig 1 displays a plastic poster which we employed to remind participants about the different steps of the activity after the explanation (steps *v* to *ix* in the list above), and also summarizes this subsection.

As a brief note on the design, an alternative version could have switched the order of steps *vi* and *vii* (i.e., completing the real-effort task before choosing between cash and vouchers for payment). Completing the real-effort task first may increase the entitlement to the earnings [34]. This may have been helpful to increase the attention paid to each decision card, and it would have probably increased the weight that loss-aversion motives would have had in the decision. Nonetheless, our concern with this alternative timing between the decisions and the task was the heterogeneous (and challenging to measure or control) responses to relative performance. Even if the payment does not depend on the time taken to separate the beans,

we feared that a correlation between performance and deservingness affected the chosen payment method. For instance, that participants who took longer in the real-effort task then selected the lower payment for reasons beyond a balance between risk exposure and non-fungibility.

## Sampling

The experiment was conducted between July 3rd and August 29th, 2021. The participants were rural subjects recruited in eleven municipalities of the Department of Quindío, located in the western central area of Colombia. Recruitment was made simultaneously with the data collection, with the help of the municipality's agricultural offices.

Quindío is the smallest mainland Colombian department by area, and it is well-known for being a coffee-growing region. Hence, the eleven selected rural municipalities have a significant participation in agricultural activities in their economy. The small size of Quindío was also helpful in making more homogeneous the transportation costs from rural workers to their respective municipality's capital.

The local offices providing technical assistance to farmers helped us recruit participants. We conducted two sessions per municipality (except in one case). Table A.1 in the S1 Appendix reports the total number of participants per municipality. Within each municipality, we conducted a session with the show-up fee in cash and another with the show-up fee in vouchers. The ascending and descending order was randomized between municipalities, and it is also reported in this table. In Salento, we conducted a single session. The reason is that this municipality has become a tourist hotspot, and farms are now more dedicated to the hospitality industry than to agriculture. The sessions took place in rooms provided by local governments and lasted between 60 and 90 minutes. On average, participants earned 37,9 kCOP for their participation.

We conducted ten additional sessions with a total of sixty-nine students in Armenia, the capital city of Quindío, in September 2021. We collected the data on this additional sample with two comparative purposes. First, students were, on average, 24.5 years old (std. dev. 6.6), and they were rarely in charge of grocery shopping. The reason is that they usually live with their elders in this city or nearby municipalities (the average number of adults in their household is 2.8, only 8.7% reported being married, and 38% reported receiving a monthly allowance). Hence, we hypothesize that students would treat the voucher as less fungible than the rural sample. Given these conditions of the students' sample, they serve as a benchmark comparison for the voucher acceptability among rural workers, regardless of $V$. Second, lab-in-the-field studies are subject to concerns about whether the protocol was clear for participants. Replicating the protocol with a students' sample is *ex ante* helpful to validate, in case of finding an unexpected behavioral pattern in the sample of interest (e.g., lack of sensibility to $V$), if it may be caused by confusion with the protocol.

The low number of student participants per session (6.9 on average) resulted from the small number of students attending in-person teaching sessions due to sanitary restrictions during the pandemic. Undergraduate students from Economics, Agriculture, and Zootechnics were invited to participate. The activity was part of one of their lectures, reducing the self-selection in the study. Moreover, the participants did not have previous experience in economic experiments, as this research methodology was not previously employed at this university. The amounts offered to participants in all decisions between cash and vouchers were the same as in the original sample, and the same team conducted the sessions. We selected a supermarket near the university to redeem the vouchers, holding the same rules as the lab-in-the-field experiment.

## Sample description

The average participant was 51 years old (std. dev. 16.9). One-third of the participants were women, and 55% at most completed elementary school. The self-reported average income was 578 kCOP, corresponding to approximately 63% of the monthly minimum wage in Colombia. Twenty-one percent reported receiving a monthly payment, whereas the rest received more frequent–but also lower–payments. Twenty-nine percent reported that part of their payment was received in kind. Regarding income, we also observe that 29% of the participants received a government subsidy. The average number of adults and children in their household was 2.8 and 0.6, respectively.

As intended, our sample captures farmers with and without land. Thirty-six percent reported having or leasing a land plot, and among them, the most common crops were coffee (52%) and plantain (18%). Regarding social security, only 22% of the participants reported being in the contributive health system. Fifty percent reported having suffered any health problem in the past 12 months. When we asked them about their financial planning for their old age, 61% reported not making any plan, 16% have a mandatory pension plan, 8% a voluntary pension plan, and 10% reported expecting help from their offspring. Informal credit networks also appear to be important in this rural context. When asked about who could lend them 50 kCOP from one day to another, 29% said a friend, 18% a neighbor, and 13% a family member. Nonetheless, 35% also replied that they do not have a person who could lend them this money.

Table A.2 (in the S1 Appendix) reports balance across treatments. Recall that variation in whether the show-up fee was paid in cash or vouchers was done between sessions. In contrast, the variation in displaying the vouchers in ascending or descending order was implemented between municipalities. As a consequence, all the descriptive variables are balanced regarding variation in the show-up fee payment, whereas the following variables are unbalanced on whether the vouchers were presented in ascending or descending order: age, marital status, education above elementary school, and whether the household belongs to the contributive or subsidized health system. These variables are included as controls in our regressions.

Regarding the students' sample, they were younger (24.5 years old, with a standard deviation 6.6) and with more female representation (52%). They reported an average monthly income of 929 kCOP. Monthly payments are also more frequent (48%), and belonging to the contributive regime was more widespread (54%). Nonetheless, the access to government subsidies was relatively similar to the field sample (20%). Fifty-seven percent reported having any health issues in the past 12 months. Fifty-one percent of students reported that their family owns or leases rural land. This is 1.41 times the frequency in the rural sample. Although it seems paradoxical at first sight, the most probable explanation is that 52% of the sample studied zootechnics, and another 26% studied a short-duration program in agrarian sciences. Only the remaining 22% studied Economics. Table A.3 in S1 Appendix reveals that all the characteristics are balanced across treatment arms.

## Results

In this section, we present the descriptive results and then proceed with the regression analyses. We report here all the treatment variations and experimental sessions we have conducted for this research question. The data and script files to replicate all the tables and figures are available in the OSF repository https://osf.io/rp4xa/.

### Voucher uptake rates

Panel (a) in Fig 2 displays, in blue, the percentage of rural participants (N = 214) that chose a voucher of value $V \in \{25; 28; 30; 34\}$ kCOP instead of a 30 kCOP cash payment. For $V = 25$,

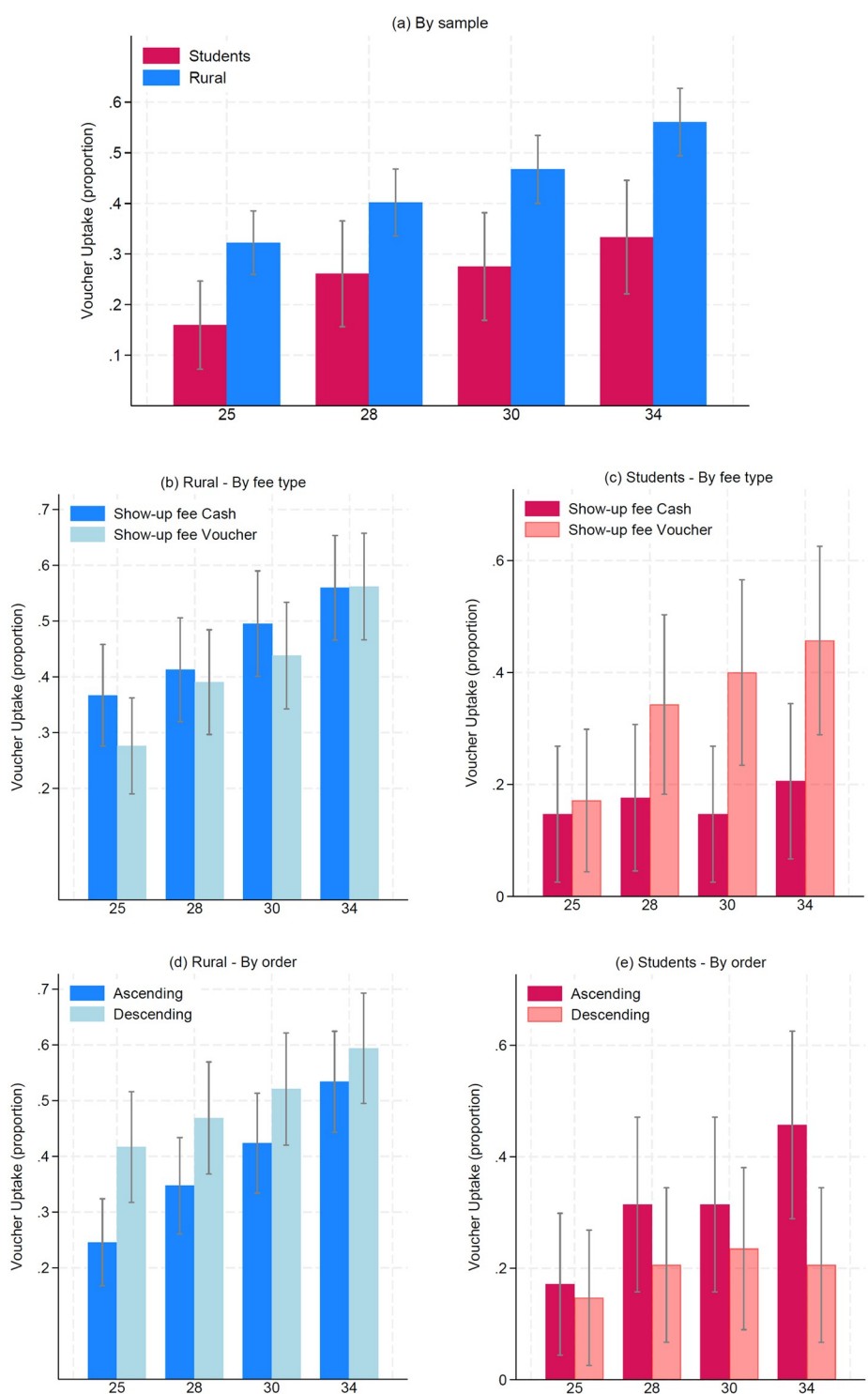

**Fig 2. Voucher uptake (as proportions) in the rural and students' sample (by treatment).**

the voucher is preferred by roughly one-third of the participants. This number increases to 41% for $V = 28$, meaning that 60% of the rural participants prefer the cash payment when both have the same expected value. For $V = 30$, less than half (47%) of participants chose a voucher that, in expected value, gave a higher payment than the cash option. In the scenario with the most generous voucher, $V = 34$, its uptake increases to 56%. Panel I in Table A.4 in S1 Appendix reports this information along with the comparisons for the rural sample shown in Panels (b) and (d) in Fig 2. Panel (b) reveals slight variation in voucher uptake depending on the type of show-up fee. This null result is confirmed by the $\chi^2$ tests reported in Table A.4 in S1 Appendix ($p$–values are 0.16, 0.74, 0.40, and 0.97 for $V$ equals 25, 28, 30, and 34, respectively). The rural workers who received a voucher for their show-up fee do not seem to perceive an additional voucher payment as more attractive, given that they already must incur a redemption cost.

Panel (d) in Fig 2 compares voucher uptake among rural participants depending on the order in which the decisions are presented. Uptake is higher when the values of $V$ are displayed in descending than in ascending order. The difference is statistically significant for $V = 25$ ($\chi^2$ test, $p$–value 0.008) and marginally significant for $V = 28$ ($p$–value 0.072), but not for $V \geq 30$ (see Table A.4 in S1 Appendix).

From a more general perspective, the uptake is increasing in $V$ for the rural workers in panels (a), (b), and (d). We interpret this pattern as a basic validation check that the instructions were properly understood.

**Comparison with the students' sample.** Panel (a) in Fig 2 reveals that voucher uptake rates are systematically lower among students (N = 69) compared to our rural sample. This result is validated through the $\chi^2$ statistical tests reported in Panel II in Table A.4 in S1 Appendix ($p$–values are below 0.01 for all the decisions, except $V = 28$ with $p = 0.035$). On top of validating that vouchers are seen as less fungible among students, Panels (c) and (e) reveal additional differences in how this sample reacted to the between-subjects conditions. Panel (c) validates the expected effect of providing the show-up fee as a voucher: it increases the attractiveness of the voucher payment because redemption costs must anyway be paid, in particular for high $V$ (the difference is statistically significant for $V$ equal to 30 and 34, with $p$–values 0.019 and 0.027). Panel (e), on the other hand, shows that students have higher voucher uptake when these are presented in ascending order, the opposite pattern of the rural sample. Nevertheless, this difference is statistically significant only for $V = 34$ ($p = 0.027$).

## Minimum acceptable voucher and main regression analysis

We define the *minimum acceptable voucher* as the lowest value of $V$ for which the participant selected the voucher instead of the cash. This aggregated measure collapses the four decisions into a single observation per participant. Our *minimum acceptable voucher* can be interpreted as a willingness to accept (the voucher instead of a cash payment), with a caveat: preference reversals (e.g., taking the voucher for $V = 28$ and $V = 34$ but the cash for $V = 30$) are treated as mistakes that do not affect the lowest switching point toward the voucher. Under this definition, we minimize the violations of fungibility, and we can run regressions using the participant (instead of the participant × decision) as the unit of analysis.

Fig 3 plots the cumulative distribution of the minimum acceptable voucher. We use a cumulative distribution given its interpretation as a willingness to accept: we obtain an idea of the aggregate voucher acceptance for each of the four values $V$ on the horizontal axis (i.e., the "steps" of the staircase). Nonetheless, we do not know the minimum voucher value for those participants who always preferred the cash (accounting for 35% of the combined samples, or 99 out of 283), so the last vertical step in the cumulative distribution is marked as *unknown*. This

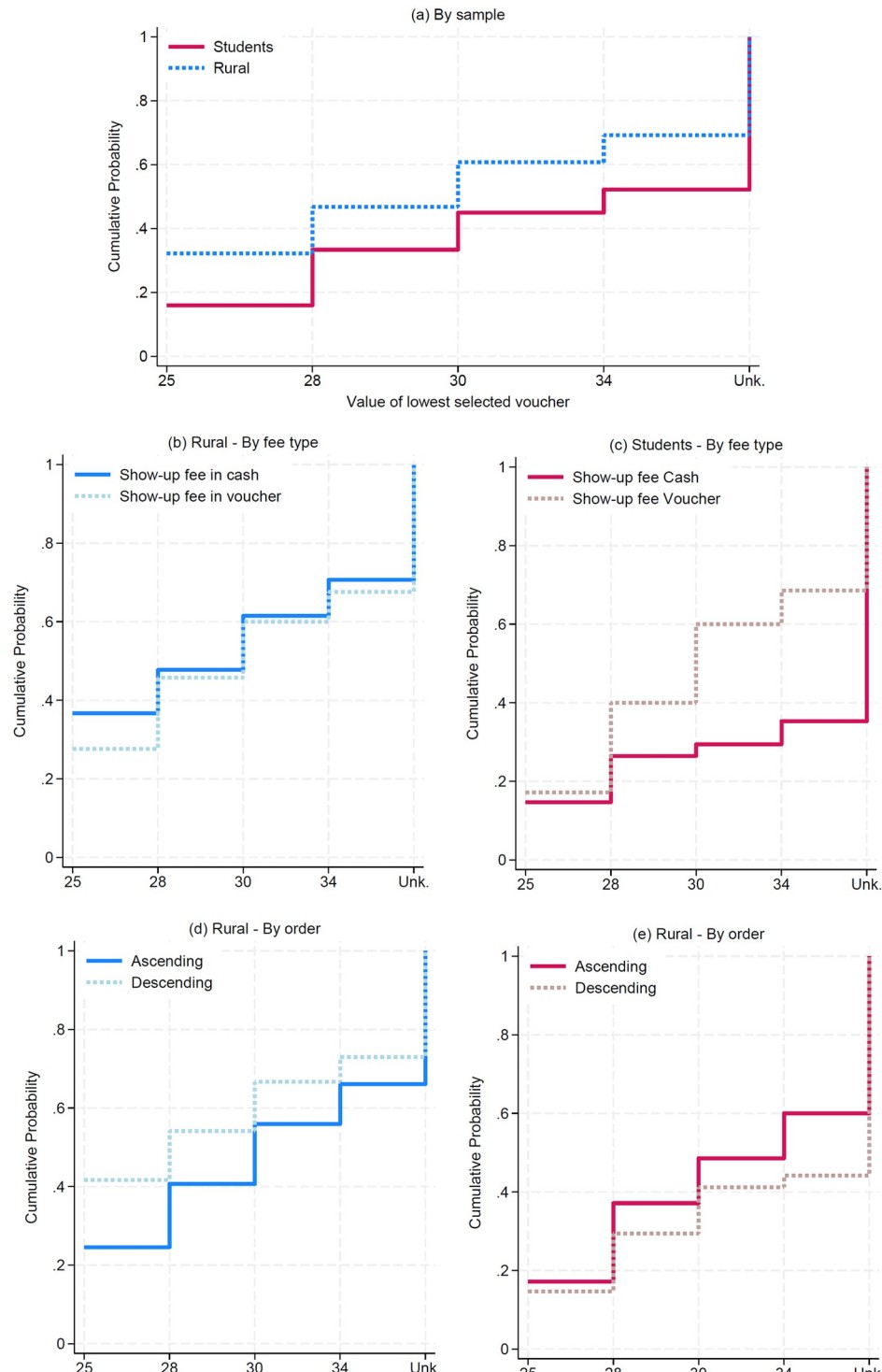

**Fig 3. Cumulative distribution of the minimum voucher value taken by each participant (by treatment).**

figure validates the comparative analyses mentioned above: the higher voucher uptake among rural workers compared to students (Panel a); the differences in voucher uptake depending on the type of show-up fee, which were relevant for students but not for rural workers (Panels b and c); and the higher attractiveness of the voucher display in descending order for rural workers, but not for students, for whom the vouchers are more attractive in ascending order.

We propose a regression analysis to understand the treatment effects on the minimum voucher price needed to make this option preferable while controlling for other observable characteristics and improving the computation of standard errors. Due to the participants who never took the voucher payment, we cannot use a simple OLS regression. Therefore, we employ two alternative econometric models that let us account for those participants who preferred the cash payment in all four decisions. First, we run a tobit regression where the dependent variable is the minimum value required to choose the voucher. Here, we assume upper-level censoring and input the value of the most generous voucher (34 kCOP) to the participants who always preferred the cash payment. Second, we run an ordered logistic regression, which is helpful for modeling monotonically increasing valuation levels without a cardinal interpretation. Here, we assign one level to each voucher value and add one final level for those participants who always preferred cash (meaning that their minimum value is above 34 kCOP).

Whereas the tobit model has a more practical interpretation, the ordered logit accounts better for the large share of participants that always preferred the cash payment. Tobit coefficients can be interpreted as in an OLS model, but the linear effect is on the uncensored latent variable (i.e., as if we would have identified minimum acceptable vouchers beyond $V = 34$), not the observed outcome. The coefficients in an ordered logit model are more convoluted to interpret because they are directly estimated as log odds. To interpret them in odds, we need to apply an exponential transformation. The positive coefficients will be transformed into odds larger than 1, indicating a higher probability of selecting highly ranked categories. By contrast, the negative coefficients will be transformed into odds between 0 and 1. In our specific case, an odd below one will indicate that an increase in the independent (explanatory) variable would reduce the selection of a higher switching valuation, meaning that vouchers are preferable as a lower $V$ is required to switch from cash to voucher payment.

Table 2 reports the coefficients for both model types. Standard errors are clustered at the randomization level, the session. We cannot include municipal fixed effects because the

**Table 2. Treatment effects on the price of accepted vouchers for rural workers.**

| | Tobit | | Ordered logit: Odd ratios | |
|---|---|---|---|---|
| | **(1)** | **(2)** | **(3)** | **(4)** |
| Show-up fee in voucher payment | 0.413 | 0.283 | 1.202 | 1.227 |
| | (1.240) | (0.745) | (0.417) | (0.255) |
| Descending order | -1.790 | -3.432*** | 0.573* | 0.404*** |
| | (1.140) | (0.854) | (0.180) | (0.105) |
| Constant | 31.78*** | 34.14 | | |
| | (1.207) | (35.08) | | |
| Observations | 214 | 214 | 214 | 214 |

Additional controls: whether the session was the first or second in the day. Municipal-level covariates included in models (2) and (4): formal employment deprivation, area, average household size, unsatisfied basic needs in the municipal seat, ratio of cultivated area over total area, average age, percentage of married population at the municipality level. The first three are statistically significant. Clustered standard errors in parentheses.

*** p<0.01,

** p<0.05,

* p<0.1.

descending order condition was randomized at the municipality level, except in one location. Instead, we include municipal-level covariates in the even-numbered models. We confirm our finding that giving the show-up fee as a voucher does not have any effect in this sample. Moreover, we validate that vouchers become more attractive when they are presented in descending order. In the tobit model, the negative coefficient indicates that a lower voucher value is needed to make participants opt for this payment option. The coefficient is negative in models (1) and (2), and it becomes larger and statistically significant after controlling for other municipality covariates. For the ordered logit, the reported coefficients in odd ratios fall below one. Hence, they indicate a lower probability of selecting a higher category (i.e., a larger switching value $V$ that makes a voucher preferable to cash). Thus, presenting the vouchers in descending order makes *less* necessary a larger voucher payment to induce this switch from cash.

One potential concern from the reader with our estimation is that our number of clusters (21) is below the recommended threshold (30). To address this, we included the sessions conducted with students (10) and added a categorical variable for participants in the field sample, which was also interacted with the two treatment variables. The results of this alternative specification, reported in Table A.5 in the S1 Appendix, are qualitatively identical. Moreover, the interaction coefficients (and their addition with the main treatment coefficients) confirm the treatment differences across samples as previously described.

## Exploratory regression analysis

We perform an additional econometric exercise to shed light on other factors affecting the voucher uptake in our field sample. This exercise has an exploratory nature. Hence, we do not have an ex-ante hypothesis on which participants' characteristics may predict the voucher uptake. We use a linear probability model with a panel data structure with each decision per participant as the unit of observation. This panel structure also lets us compute the marginal effect of increasing the voucher value on its uptake rate. We control for all the municipal-level covariates employed in the previous exercise, although we do not report their effects either. We add the individual covariates in two sets. First, we add those related to the participant's demographic characteristics. Second, we added other variables that would hint at what makes the participants more likely to treat the vouchers as fungible with cash. All the following results hold if we take the participant as the unit of analysis in a tobit model (see Table A.6 in the S1 Appendix).

Table 3 reports the results of this regression. Since the panel structure gives us four observations per participant, we clustered the standard errors at the participant level. We find that increasing the voucher value by 1 kCOP increases the uptake by 3.2 percentage points. Moreover, the previous results for the two treatment variables hold.

By looking at the covariates added to model (2), we find that participants reporting a higher willingness to take risks are more likely to choose the voucher. We employ the measure of risk-taking attitudes proposed by [33]. This result sounds counter-intuitive from a traditional risk-aversion framework, as one would expect a higher willingness to opt for the voucher among the most risk-averse participants. Among the other demographic characteristics added to model (2), we find that rural workers who reported not working their own land were less likely to opt for the voucher payment. This coefficient hints that payment substitutes for cash appear less fungible to rural workers in poorer conditions. On the other hand, we find that gender, age, educational attainment, and marital status do not predict voucher uptake.

The second set of added covariates, in model (3), reveals that participants reporting to receive any government subsidy are more likely to choose the voucher. The most commonly reported subsidies in our sample were *Familias en Acción*, a transfer conditional on sending

**Table 3. Linear probability model exploring individual predictors of voucher take-up.**

| | (1) | (2) | (3) |
|---|---|---|---|
| Voucher value | 0.0322*** | 0.0324*** | 0.0324*** |
| | (0.00549) | (0.00555) | (0.00556) |
| Show-up fee in voucher | -0.0443 | -0.0373 | -0.0398 |
| | (0.0498) | (0.0496) | (0.0493) |
| Descending order | 0.509** | 0.479* | 0.471* |
| | (0.236) | (0.244) | (0.245) |
| Voucher value × Descending order | -0.0122 | -0.0122 | -0.0122 |
| | (0.00766) | (0.00775) | (0.00776) |
| Willingness to take risks | | 0.0191* | 0.0233** |
| | | (0.00977) | (0.00977) |
| Agricultural laborer | | -0.151** | -0.132** |
| | | (0.0633) | (0.0648) |
| Women | | 0.0450 | 0.0298 |
| | | (0.0596) | (0.0638) |
| Age | | 0.00143 | 0.000791 |
| | | (0.00201) | (0.00201) |
| Primary school or less | | -0.0441 | -0.0372 |
| | | (0.0654) | (0.0664) |
| Married | | -0.0213 | -0.0141 |
| | | (0.0691) | (0.0668) |
| Government's subsidy | | | 0.0994* |
| | | | (0.0571) |
| Monthly salary | | | 0.107 |
| | | | (0.0708) |
| Payment in-kind | | | -0.0243 |
| | | | (0.0555) |
| Constant | -0.171 | 0.832 | 2.110 |
| | (2.682) | (2.739) | (2.805) |
| Observations | 856 | 848 | 848 |
| R-squared | 0.091 | 0.133 | 0.146 |

Municipality-level covariates in all regressions: formal employment deprivation, area, average household size, unsatisfied basic needs in the municipal seat, ratio of cultivated area over total area, average age, percentage of married population at the municipality level. The definition of agricultural laborer is a worker who does not own or leases land. Standard errors with clusters at the individual level in parentheses.

*** p<0.01,

** p<0.05,

* p<0.1.

their kids to school; and *Colombia Mayor*, targeted to the elderly population. Since both subsidies are delivered in cash, we rule out that this finding, implying more fungibility among those receiving subsidies, is driven by being accustomed to in-kind transfers.

## Discussion

### What do we learn by comparing rural workers with students?

Recall from Figs 2 and 3 that vouchers were more appealing to rural workers (32 to 56% of uptake) than to students (16 to 33% of uptake). Since the two most valuable vouchers ($V \in 28$,

32) offered a higher expected value and lower risk exposure than the cash payment, these results suggest an intermediate acceptability among rural workers, even if the vouchers are subsidized. The comparison with students must be made with caution, since it is more helpful to ratify and put in perspective the results for the field sample than for reaching direct conclusions about the fungibility assumption among students. The main issue is that the voucher payment can be distortionary among students not directly in charge of their household's food expenditure. Hence, we cannot argue that fungibility is less likely to hold among students.

With this caveat in mind, we learned two lessons from the between-sample comparison. First, rural workers understood and valued the risk reduction that accompanied the voucher payment. In addition to the higher (and statistically different) voucher uptake rate for every $V$, the order effects suggest that the reference points are driven by different attributes in each sample. Rural workers took the vouchers more often when they were presented in descending order. In line with the hypothesis described at the end of the Experimental Design section, one explanation for this result is that the partial insurance from the voucher payment appears to trigger an endowment effect that maintains the attractiveness of this payment method in the subsequent decisions with lower voucher values. In other words, rural workers assign more value to the voucher because of the implicit insurance it brings. Hence, when the value of the voucher decreases in further decisions, it is less likely that these workers will switch to the cash option because they value more the implicit risk reduction compared to the participants who took the decisions in ascending order and did not benefit from the lowest risk exposure. This endowment effect for risk has been documented, with an emphasis on the differences in the predicted behavior when the reference point is stochastic [35]. More importantly, lab-in-the-field evidence from Uganda reports a similar pattern when tokens allocated on a lottery are by default on the safe or risky choice [36]: more risky starting points yield less risk aversion in an identical choice set. In our setting, those who started seeing the insurance as more attractive, and have a less risky starting point, have a higher valuation of the risk reduction and are implicitly behaving as more risk-averse.

Recall also that this pattern was not present among students. For them, the uptake rates were slightly larger (and non-significant) when the vouchers were displayed in ascending order. We conjecture that in their case, the reference point was the voucher's value instead of the risk reduction, making the voucher rapidly unattractive as it decreased in value in the descending order condition (see panel (e) in Fig 2).

Second, we validate that voucher redemption costs are important when the participants in our study have not planned any food expenditure. In our rural sample, we did not find an effect of the show-up fee payment method, presumably because most participants planned to incur such costs during their visit to the municipality's capital. Hence, the transaction costs we randomly induced were negligible compared to their total travel costs of visiting the municipality's capital to make these purchases. By contrast, the students were surprised by the invitation to participate in our study while attending class. If they did not plan to make any food expenditure in a supermarket, the difference in transaction costs across show-up fee payment methods was more critical in their decision.

## Comments on the exploratory analysis

Table 3 sheds light on two patterns connecting the acceptability of voucher payments with reported variables in the survey, for which we expand our discussion.

First, participants reporting a higher willingness to take risks are more likely to choose the voucher. We offer two conjectures for this result. On the one hand, more risk-averse participants may perceive the voucher as less fungible if they see the cash payment as some

unexpected earnings that can be saved to protect against future shocks. This behavior is consistent with the evidence on how savings can increase among vulnerable populations through a mental accounting effect [37]. On the other hand, participants may be engaging in risk compensation behavior, defined as an adjustment in risk exposure in response to the perceived diminishing level of risk [38, 39].

The second pattern is that the voucher appears to be less attractive (or less acceptable) among landless agricultural workers (i.e., those who do not work their owned or leased land, commonly known as *peones* or *jornaleros*). We find that being a landless worker is positively correlated with reporting to receive in-kind payments (Spearman's $\rho = 0.34$, $p < 0.001$), so this does not seem to be explained by them not being used to payment methods different from cash. A second alternative is that landless workers are also more vulnerable, and their opportunity cost of forgoing a cash payment is much higher (e.g., for the mental accountability reasons mentioned above). Our evidence is non-conclusive in this aspect: although there is a negative correlation between being a landless worker and the logarithm of income (Spearman's $\rho = -0.18$, $p = 0.009$), Table 3 showed that those receiving subsidies are more likely to accept the voucher.

## Concluding remarks

We conducted a lab-in-the-field experiment to determine whether Colombian rural workers are willing to accept a voucher payment, accompanied by a risk reduction, instead of a cash payment. We find that 56% opted for the voucher when it has an expected value 23% higher than cash and offers lower risk exposure. It implies that 44% of our participants perceive that the additional expected payment and lower risk are insufficient to overcome the voucher's non-fungibility. In the least generous of the four decisions that participants took, when the voucher's expected value was 12% lower than the cash payment, only one in three participants chose the voucher. Nonetheless, these uptake rates are consistently larger compared to a sample of university students from the same region. We conclude that the risk reduction accompanying the voucher payment was at least mildly attractive to rural workers.

The attractiveness of the risk reduction is reinforced by one of our treatment outcomes: presenting the vouchers in descending order yields a higher uptake than the ascending order. We interpret this result in the light of an endowment effect for the insurance: the implicit risk reduction of taking valuable vouchers reduces the likelihood of opting for the cash payment when the voucher value decreases because this insurance becomes costly to forgo. Similar findings on the ordering of choices and uptake of insurance exist in the health sector [40, 41], and they call for choice architecture interventions to help people make more affordable insurance decisions. We extend the exploration of ordering to the rural sector in developing countries. This is a policy-relevant discussion where our results suggest that the cost of non-fungibility perceived by rural workers can be diminished if the risk-pooling mechanisms are sufficiently attractive and, perhaps, initially subsidized to increase the risk share that can be transferred to the cooperative from their members.

Our exploratory analysis reveals that landless rural workers tend to see the voucher as less valuable with respect to (even lower amounts of) cash. This result has a dismal interpretation: rural workers without land as collateral appear more reluctant to opt for risk-reducing alternatives when their cost is a non-fungible payment. This pattern may reinforce the negative and permanent effects that shocks may have in the medium- and long-run among these workers, so the need to find alternative insurance mechanisms that may look attractive remains.

## Supporting information

**S1 Appendix. Supplementary tables, figures and experimental protocol.** They can be found in the corresponding link.
(PDF)

## Acknowledgments

We gratefully acknowledge the research assistance provided by Laura Acosta Bedoya, Julián Cárdenas Ospina, Merian Chica Otálvaro, Michael Marín Piedrahita, Juliana Olarte Zuluaga, José Santiago Rodas Vélez, Kelly Torres Hurtado, and Leidy Torres Quintero. We appreciate the comments provided by Andrew Foster, Margarita Gáfaro, Karen Macours, and the participants at the Friends of IAST Conference, EfD Annual Meeting, and *Congreso de la Red de Investigadores* from the Colombian Central Bank.

## Author Contributions

**Conceptualization:** Alexander Cano, Darwin Cortés, César Mantilla, Medardo Restrepo.

**Data curation:** Alexander Cano, Laura Prada.

**Funding acquisition:** Darwin Cortés, César Mantilla, Medardo Restrepo.

**Methodology:** Darwin Cortés, César Mantilla, Laura Prada.

**Project administration:** Alexander Cano, Laura Prada, Medardo Restrepo.

**Resources:** Alexander Cano.

**Supervision:** Alexander Cano, César Mantilla, Medardo Restrepo.

**Validation:** Alexander Cano, Darwin Cortés.

**Visualization:** César Mantilla, Laura Prada.

**Writing – original draft:** César Mantilla, Laura Prada.

**Writing – review & editing:** César Mantilla.

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
