## [Decision Letter · Decision Letter 0]

10 Oct 2023

PONE-D-23-23026Non-fungibility reduces the attractiveness of risk-reducing payments among rural workers: a lab-in-the-field experiment in ColombiaPLOS ONE

Dear Dr. Mantilla,

Thank you for submitting your manuscript to PLOS ONE. After careful consideration, we feel that it has merit but does not fully meet PLOS ONE’s publication criteria as it currently stands. Therefore, we invite you to submit a revised version of the manuscript that addresses the points raised during the review process.

We look forward to receiving your revised manuscript.

Kind regards,

Kenju Akai, Ph.D.Economics

Academic Editor

PLOS ONE

Journal Requirements:

 "DC, CM and MR received funding from MinCiencias and the World Bank through Alianza  EFI. Grant number: FP44842-220-2018"

"Financial Support from the program “Inclusi´on productiva y social: programas y pol´ıticas para la promoci´on de una econom´ıa formal, c´odigo 60185, que

conforma la Alianza EFI, bajo el Contrato de Recuperaci´on Contingente No. FP44842-220-2018.”

is gratefully acknowledged."

"DC, CM and MR received funding from MinCiencias and the World Bank through Alianza  EFI. Grant number: FP44842-220-2018"

7. Please amend your list of authors on the manuscript to ensure that each author is linked to an affiliation. Authors’ affiliations should reflect the institution where the work was done (if authors moved subsequently, you can also list the new affiliation stating “current affiliation:….” as necessary).

8. We note that Figure 1 in your submission contain copyrighted images. All PLOS content is published under the Creative Commons Attribution License (CC BY 4.0), which means that the manuscript, images, and Supporting Information files will be freely available online, and any third party is permitted to access, download, copy, distribute, and use these materials in any way, even commercially, with proper attribution. For more information, see our copyright guidelines: http://journals.plos.org/plosone/s/licenses-and-copyright.

Additional Editor Comments:

The experimental design is carefully thought out and well sampled. The study is interesting and worthy of being an experiment to test financial tolerance in developing countries. However, for the benefit of the reader, please add the following points

1) Please detail the progression procedure step-by-step, as it may be misleading for MLPs. Also, please explain in detail how to determine the show-up fee.

2) Why is the expected utility hypothesis not valid? Why can't people's behavior during the experiment be explained by the certainty equivalent in expected utility as presented in the classical insurance model? Discuss this in detail. Or use prior research to explain this in your intro.

Reviewers' comments:

Reviewer's Responses to Questions

**Comments to the Author**

1. Is the manuscript technically sound, and do the data support the conclusions?

Reviewer #1: Yes

Reviewer #2: Yes

2. Has the statistical analysis been performed appropriately and rigorously? 

Reviewer #1: Yes

Reviewer #2: Yes

3. Have the authors made all data underlying the findings in their manuscript fully available?

Reviewer #1: Yes

Reviewer #2: Yes

4. Is the manuscript presented in an intelligible fashion and written in standard English?

Reviewer #1: Yes

Reviewer #2: Yes

5. Review Comments to the Author

Reviewer #1: "Non-fungibility reduces the attractiveness of risk-reducing payments among rural workers: a lab-in-the-field experiment in Colombia."

1. The title is descriptive but might be too lengthy for quick comprehension. Consider shortening the title while retaining its essence, e.g., "Non-fungibility and Risk Preferences: A Field Experiment in Colombia."

2. The abstract provides a clear overview but lacks specific results or statistical significance indicators. Include key statistical findings in the abstract to give readers immediate insight into the study's outcomes.

3. While the choice between cash and voucher is clear, the rationale behind the specific voucher values and their expected outcomes is not elaborated upon. Provide a theoretical framework or prior research that guided the choice of voucher values. Discuss the expected behaviour based on economic or psychological theories.

4. The comparison with undergrad students might not be directly relevant or might lack context. Elaborate on the reason for this specific comparison. If it's to show a difference in risk preferences based on demographics, then more demographic data should be provided.

5. The finding about the order of voucher presentation is interesting but lacks a deeper exploration. Delve into the psychological or behavioural reasons behind this finding. Consider referencing literature on choice architecture or framing effects.

6. The source of funding is disclosed, but there's no discussion on how it might have influenced the research. Include a statement about the independence of the research from the funding sources.

7. Consent was obtained, but there's no mention of the process or any potential risks conveyed to participants. Elaborate on the ethical considerations, especially how participants were informed about the study's purpose and potential risks.

8. The statement about data availability is generic. Specify where the data can be accessed, whether it's an online repository or upon request, and ensure it's anonymized.

9. Your study on the behavior of rural workers in Colombia provides valuable insights into the dynamics of rural labor markets. To further enrich your analysis and provide a broader context, I recommend referring to the following studies that delve into various aspects of rural labor dynamics in different regions:

• Zeeshan, Mohapatra, G., & Giri, A. K. (2022). How Farm Household Spends Their Non-farm Incomes in Rural India? Evidence from Longitudinal Data. The European Journal of Development Research, 34(4), 1967-1996.

• Santana, R. R. C., & Ristum, M. (2023). Child Labor in Families of Rural Workers: The Issue of Intergenerationality. Trends in Psychology, 1-16.

• Kolié, D., Van De Pas, R., Codjia, L., & Zurn, P. (2023). Increasing the availability of health workers in rural sub-Saharan Africa: a scoping review of rural pipeline programmes. Human Resources for Health, 21(1), 20.

Final Recommendation:

Given the current state of the paper, I would recommend a "Revise and Resubmit." The research topic is relevant, and the methodology is sound. However, addressing the above critiques will provide a more comprehensive and insightful paper.

Reviewer #2: The manuscript “Non-fungibility reduces the attractiveness of risk-reducing payments

among rural workers: a lab-in-the-field experiment in Colombia” presents results of a series of field-based economic experiments to understand how rural people in Colombia value different payment types. The results are interesting, and mostly easy to understand. The quality of the writing is excellent, something that I greatly appreciate as it makes my work as a reviewer much easier. There are only small grammatical errors, some (but not all) of which I have flagged below. Thanks also for providing the version with line numbers.

I should say at this point that while I do use field experiments in rural areas of low- and middle-income countries, I do so mostly to test practical aspects of different incentive program implementations. My own research does not aim to test basic economic theory, something I have only basic familiarity with. When agreeing to review this paper I had not realized how much it was focused on fundamental theory, and I hope that at least one other reviewer of this paper is a real economist! I also note that some of the preferences I express in this review probably reflect that I come from a different disciplinary background.

Incidentally, I once did a pilot study in the same region you worked in (specifically the town of Filandia), with a demographically similar participant pool, and was impressed at how good at math they were. Perhaps it reflects particularly good instruction at schools there, but I’ve never seen rural workers anywhere else who figured out the optimal solutions to the game we were using so quickly. Thus, I would guess that many of your participants would have had a good grasp on the expected value of the different payment options. This probably has bearing on your results, particularly on where else they may be more or less relevant.

-Carl Salk

Specific comments:

Line 7: “developing context” – is a word missing here? Developing country?

L41-65: This seems to be a bit too much results for so early in the intro. Perhaps summarize this as a few sentences at the very end of the intro?

L55: Is a word missing after “incur”?

L92: Please also provide the approximate value of the payout in US dollars.

L126: Change to “Panel A in Table 1 summarizes the comparisons...”

L165: Perhaps I am missing something, but I thought there were only two choices (i.e. those shown from L92-95). Please clarify! Ok, on going back to L100, I think I understand this now – it seems that participants were asked to make 4 different decisions. This could made more clear up front.

L168: “play” → “plays”

L193 “draw” → “drew”

L196 “explain” → “explained.” There is a lot of switching between present and past tense in this section. I would recommend past tense, but whatever you choose, please be consistent!

L199: Delete “on”

L261-262: Change “Quindío is the smallest Department in total extension” to “ Quindío is the smallest department by area.”

L282 (and 285): “University” does not need to be capitalized since you are not using the word as part of the specific name of the university here.

L288: The first sentence here is unnecessary.

L294: Change “government’s” to “government”

L295: “were” → “was”

L325-7: Names of the fields of study do not need to be capitalized.

L333, etc.: Percentages can generally be written out as numbers.

L337-340 (and many other similar passages): Move to discussion section.

L347 (and elsewhere): Just need to say “incur” rather than “incur in.”

Table 1: I realize tables like these are standard for economists, but in many other disciplines a series of barplots would be used, something I generally find easier to interpret.

L382: Should say “participants.”

Figure 2: Again this is probably a disciplinary difference, but the stair-stepped graphs are odd to me – I would prefer a standard line graph with individual data points shown using symbols. This would also reduce the overlap of the two lines. Also, I find the >34 category always being 100% to be a bit misleading (and hard to even see as presented since the black dashed lines always disappear behind the gray lines – if you want to keep the stair-stepped look, you could at least make the black dashed lines thicker so they stick out behind the gray lines). There may be participants who always prefer cash, no matter how large the voucher value is.

Table 2: This table is confusing in that (and I’m still not 100% certain about this) a negative coefficient in the tobit analysis indicates the same thing that a value between 0 and 1 means for the ordered logit analysis. At first glance, I thought you had different results since one set of coefficients was negative and the other positive, but then I noticed that one was an odds ratio. This difference needs to be highlighted to prevent others from getting confused like I did.

L452: Change “neither” to “either.”

L472-3: Change “incurring” to “engaging.”

L487: Change “worst” to “worse.”

6. PLOS authors have the option to publish the peer review history of their article (what does this mean?). If published, this will include your full peer review and any attached files.

Reviewer #1: No

Reviewer #2: **Yes: **Carl Salk

---

## [Author Response · Author response to Decision Letter 0]

10 Nov 2023

We have uploaded the responses as the additional file "Response letter_RR.pdf".

---

## [Decision Letter · Decision Letter 1]

8 Dec 2023

PONE-D-23-23026R1Are non-fungible payments attractive when they reduce risk exposure? Evidence from ColombiaPLOS ONE

Dear Dr. Mantilla,

Thank you for submitting your manuscript to PLOS ONE. After careful consideration, we feel that it has merit but does not fully meet PLOS ONE’s publication criteria as it currently stands. Therefore, we invite you to submit a revised version of the manuscript that addresses the points raised during the review process.

We look forward to receiving your revised manuscript.

Kind regards,

Kenju Akai, Ph.D.Economics

Academic Editor

PLOS ONE

Journal Requirements:

Additional Editor Comments :

Unfortunately, I lost contact with the first reviewer, but as the editor, it was my responsibility to peer review and make sure there were no problems.

As the editor, your response to the review was accurate. One last very minor request.

Please consider it if you can.

If you think this is irrelevant or not at all, please reply to that effect.

It will not cause us to reject your paper.

Your manuscript is almost accepted already.

Here are some additional questions related to the points the first reviewer questioned.

Although these two papers or studies were not referenced,

Can these studies be related to your study?

If you can, please briefly mention them in either the introduction or the discussion.

Holt, Charles, A., and Susan K. Laury. 2002. "Risk Aversion and Incentive Effects ." American Economic Review, 92 (5): 1644-1655.

DOI: 10.1257/000282802762024700

Tanaka, T., Camerer, C. F., & Nguyen, Q. (2010). Risk and time preferences: Linking experimental and household survey data from Vietnam. American economic review, 100(1), 557-571.

Reviewers' comments:

Reviewer's Responses to Questions

**Comments to the Author**

1. If the authors have adequately addressed your comments raised in a previous round of review and you feel that this manuscript is now acceptable for publication, you may indicate that here to bypass the “Comments to the Author” section, enter your conflict of interest statement in the “Confidential to Editor” section, and submit your "Accept" recommendation.

Reviewer #2: (No Response)

2. Is the manuscript technically sound, and do the data support the conclusions?

Reviewer #2: Yes

3. Has the statistical analysis been performed appropriately and rigorously? 

Reviewer #2: Yes

4. Have the authors made all data underlying the findings in their manuscript fully available?

Reviewer #2: Yes

5. Is the manuscript presented in an intelligible fashion and written in standard English?

Reviewer #2: Yes

6. Review Comments to the Author

Reviewer #2: Thank you for your revisions. This manuscript is now in excellent shape and I enthusiastically recommend its acceptance. Below I have noted some minor points that could further improve the clarity of the writing.

Best regards,

Carl Salk

Abstract (and elsewhere): The word “uptake” would sound more natural than “take-up.”

L85: Change to “developing-country context”

L138: Change “less” to “least.”

L186-187: Change to “the municipalities’ capitals tend to be relatively compact.” In general (and I can’t believe I didn’t catch this in the first review), the phrase “municipality’s head” sounds like a person who is the chief executive of the municipality. I think you are talking about the town where the municipal government is based. There is probably no perfect translation for this in English, but “capital” is my best bet. “Market town” or “main town” may also work. I note this phrase appears several times in the manuscript.

L330: I think “list” would be clearer than “timing” here.

L350: “Quindio is the smallest mainland Colombian department by area.”

L380: “In-person” would sound more natural than “in-presence.”

L388: Change “of” to “as.”

L404: Change “during” to “for.”

Figure 2: I think the y-axes should be labeled “proportion” rather than “%.” Or the numbers on the scale could be multiplied by 100 to fix this.

L485: I believe this should say “Figure 3,” not “Figure 2.”

L487: “On” would sound more natural than “in.”

L560: Change “make” to “makes.”

L562: “Tobit” does not need to be capitalized.

L565: Change “in” to “by.”

7. PLOS authors have the option to publish the peer review history of their article (what does this mean?). If published, this will include your full peer review and any attached files.

Reviewer #2: **Yes: **Carl F Salk

---

## [Author Response · Author response to Decision Letter 1]

8 Dec 2023

See attached the response letter.

---

## [Editor Report · Decision Letter 2]

10 Dec 2023

Are non-fungible payments attractive when they reduce risk exposure? Evidence from Colombia

PONE-D-23-23026R2

Dear Dr. Mantilla,

We’re pleased to inform you that your manuscript has been judged scientifically suitable for publication and will be formally accepted for publication once it meets all outstanding technical requirements.

Kind regards,

Kenju Akai, Ph.D.Economics

Academic Editor

PLOS ONE

Additional Editor Comments (optional):

This is the best experimental paper ever I have read in this journal.
---

## [Editor Report · Acceptance letter]

18 Dec 2023

PONE-D-23-23026R2 

PLOS ONE

Dear Dr. Mantilla, 

I'm pleased to inform you that your manuscript has been deemed suitable for publication in PLOS ONE. Congratulations! Your manuscript is now being handed over to our production team.

Kind regards, 

on behalf of

Dr. Kenju Akai 

Academic Editor

PLOS ONE